# Stable Production of a Recombinant Single-Chain Eel Follicle-Stimulating Hormone Analog in CHO DG44 Cells

**DOI:** 10.3390/ijms25137282

**Published:** 2024-07-02

**Authors:** Munkhzaya Byambaragchaa, Sei Hyen Park, Sang-Gwon Kim, Min Gyu Shin, Shin-Kwon Kim, Myung-Hum Park, Myung-Hwa Kang, Kwan-Sik Min

**Affiliations:** 1Carbon-Neutral Resources Research Center, Hankyong National University, Anseong 17579, Republic of Korea; munkhzaya@hknu.ac.kr; 2Institute of Genetic Engineering, Hankyong National University, Anseong 17579, Republic of Korea; 3Graduate School of Animal Biosciences, Hankyong National University, Anseong 17579, Republic of Korea; mrtree119@naver.com (S.H.P.); tom0391@naver.com (S.-G.K.); 4Aquaculture Research Division, National Institute of Fisheries Science, Busan 46083, Republic of Korea; smg159@korea.kr (M.G.S.); ksk4116@korea.kr (S.-K.K.); 5TNT Research, Sejong 30141, Republic of Korea; pmh@tntresearch.co.kr; 6Department of Food Science and Nutrition, Hoseo University, Asan 31499, Republic of Korea; mhkang@hoseo.edu; 7Division of Animal BioScience, School of Animal Life Convergence Sciences, Institute of Genetic Engineering, Hankyong National University, Anseong 17579, Republic of Korea

**Keywords:** rec-eel FSH analog, stable expression from CHO DG44 cells, cAMP response, pERK1/2 analysis

## Abstract

This study aimed to produce single-chain recombinant Anguillid eel follicle-stimulating hormone (rec-eel FSH) analogs with high activity in Cricetulus griseus ovary DG44 (CHO DG44) cells. We recently reported that an O-linked glycosylated carboxyl-terminal peptide (CTP) of the equine chorionic gonadotropin (eCG) β-subunit contributes to high activity and time-dependent secretion in mammalian cells. We constructed a mutant (FSH-M), in which a linker including the eCG β-subunit CTP region (amino acids 115–149) was inserted between the β-subunit and α-subunit of wild-type single-chain eel FSH (FSH-wt). Plasmids containing eel FSH-wt and eel FSH-M were transfected into CHO DG44 cells, and single cells expressing each protein were isolated from 10 and 7 clones. Secretion increased gradually during the cultivation period and peaked at 4000–5000 ng/mL on day 9. The molecular weight of eel FSH-wt was 34–40 kDa, whereas that of eel FSH-M increased substantially, with two bands at 39–46 kDa. Treatment with PNGase F to remove the *N* glycosylation sites decreased the molecular weight remarkably to approximately 8 kDa. The EC_50_ value and maximal responsiveness of eel FSH-M were approximately 1.23- and 1.06-fold higher than those of eel FSH-wt, indicating that the mutant showed slightly higher biological activity. Phosphorylated extracellular-regulated kinase (pERK1/2) activation exhibited a sharp peak at 5 min, followed by a rapid decline. These findings indicate that the new rec-eel FSH molecule with the eCG β-subunit CTP linker shows potent activity and could be produced in massive quantities using the stable CHO DG44 cell system.

## 1. Introduction

Follicle-stimulating hormone (FSH), a member of the glycoprotein hormone family, is composed of a common α-subunit and specific β-subunit in the same species [1,2]. These glycoproteins are secreted from the pituitary gland and control gonadal function in mammals and fish [3,4]. The α- and β-subunits comprise noncovalently linked subunits, and eel FSH α- and β-subunits are composed of 93 and 105 amino acids, respectively [5]. The α-subunit has two glycosylation sites at Asn^56^ and Asn^79^, and the β-subunit includes Asn^5^ and Asn^22^, demonstrating that it is post-translationally modified. Differences in glycosylation have a considerable influence on the half-life and biological activity [2,6]. In eel glycoproteins, including FSH and luteinizing hormone (LH), the specific glycosylation sites are important determinants of signal transduction through these receptors; in particular, the α-subunit Asn^56^ glycosylation site in eel LH and FSH is indispensable for biological activity [5,7]. We have previously reported that the oligosaccharide chains in LH, FSH, and equine chorionic gonadotropin (eCG) are very important for cAMP/PKA signal transduction through these receptors [1,8,9]. 

The production of recombinant (rec) hormone in the Cricetulus griseus ovary-K1 (CHO-K1) cells is very limited with respect to quantity [5,7,10]. We have also reported the production of rec-eel LH and FSH proteins in the *Bombyx mori* system; however, glycosylated proteins do not exhibit good biological activity when using this system, despite large quantities of rec-proteins [11]. Recently, many studies of CHO-DG44 suspension cells have demonstrated the large-scale production of rec-therapeutic proteins, including erythropoietin (119 mg/L) in a shake flask [12], human α-thrombin (1.5 g/L) [13], human dihydrofolate reductase (DHFR)-deficient CHO-DG44 cells [14], human alpha-1 antitrypsin (1.05 g/L) by methotrexate (MTX) amplification [15], human growth hormone (hGH) [16], and hFSH in CHO DG44 cells/CHO dhfr cells (30–45 IU/L) [17]. 

Previous studies have reported that attachment of the hCG β-subunit carboxy-terminal peptide (CTP) linker substantially increases the in vivo potency and circulatory half-lives of the hFSH β-subunit [18], hCG α-subunit [19], TSH β-subunit [20], and erythropoietin [21]. The fusion of the eCG β-subunit CTP linker to the N-terminus, C-terminus, and between the β-subunit and α-subunit in tethered Anguillid eel LH and eel FSH affects early secretion and signal transduction [22]. Therefore, *N*-linked oligosaccharides are critical for receptor binding and bioactivity of the glycoprotein hormones. A study has shown that *O*-linked oligosaccharides are not important for in vitro bioactivity or receptor binding; however, they contribute in vivo to bioactivity and half-life [23]. Various studies and related clinical trials on the addition of hCG β-subunit CTP have provided further insights on the hGH MOD-4023. One copy of the hCG β-subunit CTP linker (117–145 aa) was fused to the 5′ end of hGH, and two copies of the CTP linker were fused to the 3′ end of hGH [24,25]. This substance was ultimately reported to be successful in a safety and dose-finding study in clinical trials for growth-hormone-deficient (GHD) children [26]. 

Glycoprotein hormone receptors, including the luteotropin/chorionic gonadotropin hormone receptor (LH/CGR) and follicle-stimulating hormone receptor (FSHR), belong to the superfamily of G protein-coupled receptors (GPCRs) [27,28]. LH/CGR and FSHR are associated with various acute and classical receptor-mediated responses, including cAMP accumulation, steroidogenesis, and phosphorylation of extracellular signal-regulated kinase (pERK1/2) activation [29,30]. hCG induces the phosphorylation of ERK1/2 in Leydig cells via two distinct and independent pathways [31]. The first pathway is entirely intracellular and relies on the cAMP/PKA pathway. The second one is independent of the cAMP/PKA pathway. The Gαs proteins and β-arrestins contribute to overall ERK signaling via two distinct pathways: a rapid and transient G protein-dependent phosphorylation of ERK and a slower but sustained β-arrestin 2-dependent activation [32,33,34,35,36]. ERK activation by the FSHR depends on both β-arrestin 1 and β-arrestin 2, indicating that one β-arrestin cannot replace the other. The β-arrestins and GPCR kinases (GRK) are involved in FSHR signal transduction [37,38].

Japanese eel is one of the most specific aquaculture fish species in East Asian countries, including Japan and Korea. However, gonadal development can be induced by the administration of commercial gonadotropin reagents such as salmon pituitary extract [4,39,40]. Therefore, this study aims to establish a mass production system to induce sex maturation using rec-eel FSH hormones.

Our previous studies demonstrated that many glycoproteins cannot be produced on a large scale in CHO-K1 and CHO-Suspension (CHO-S) cells [2,5,6,10,22]. Thus, we chose the CHO DG44 cells to stably produce rec-eel FSH-wt and FSH-M. The eCG β-subunit CTP linker with 12 *O*-linked glycosylation sites was fused between the β-subunit and α-subunit of tethered eel FSH-wt to improve the biological activity and extend the half-life. This potent new eel FSH is beneficial for artificial ovulation, indicating that it has potent biological activities. The new rec-eel FSH-M protein could be produced continuously in the CHO DG-44 cell system. 

## 2. Results 

### 2.1. Isolation of Single Cells after Transfection

After transfection into CHO DG44 cells, the cells were cultured in CD-OptiCHO medium for approximately 3 weeks. The cell viability gradually decreased to 46% by the fourth medium exchange and then increased continuously to over 95% during the next three medium exchanges. The selected cells were stored in a LN_2_ tank for each cell pool. In the second round of MTX selection, the viability decreased to 26–32% after treatment with 500 nM MTX for 3 weeks. An increase in the MTX concentration (2 and 4 µM) did not decrease the viability to <80%, demonstrating that the cells slightly decreased and then recovered to over 90% within 2 weeks. MTX-selected cells were isolated using a complete cloning medium in 96-well plates. Cell clones of eel FSH-wt were observed under a microscope pre-isolation on a 96-well plate (Figure 1). Ten monoclonal cells expressing rec-eel FSH-wt were isolated and stored in a LN_2_ tank. Subsequently, an experiment was conducted to measure the production of rec-FSH proteins by culture day.

### 2.2. Western Blotting for Single-Colony eel FSH-wt Detection

The molecular weight of rec-eel FSH-wt was determined by Western blotting using an anti-eel FSH monoclonal antibody (5A14). Each cell clone was cultured, and the supernatant was collected on day 9 post-culture. Twenty microliters of the medium were used for Western blotting. Specific bands were distinctly detected at 34–40 kDa for all samples, with strong band intensity (Figure 2A). Rec-eel FSH-wt was secreted in large quantities into the culture medium, and the secretion pattern was determined according to culture time for four clones (3, 5, 8, and 9). Western blotting revealed wide bands (Figure 2B). Weak bands were detected on day 3 post-culture in all clones. The band intensity increased steadily, with the strongest signal detected on day 9, and then decreased slightly on day 11. Sodium dodecyl sulfate-polyacrylamide gel electrophoresis (SDS-PAGE) was performed on a 12% reducing gel, followed by Coomassie Brilliant Blue staining. The culture medium (5, 10, and 20 μL) of the tethered eel FSH-wt and FSH-M was applied to the SDS-PAGE analysis, respectively (Appendix A). The western blot raw date of rec-eel FSH-wt were shown at Appendix A.

### 2.3. Secretion Quantity of rec-eel FSH-wt Protein

To analyze the quantity secreted into the cell medium, the supernatant was collected on days 1, 3, 5, 7, 9, and 11 post-culture. The concentration of the rec-eel FSH-wt protein increased gradually until the final supernatant was collected, as shown in Figure 3. Rec-FSH-wt concentration was slightly detected from 553 to 1777 ng/mL on day 3 and increased to 1622–2690 ng/mL on day 5. The concentrations in the four clones were high (i.e., 2802 ± 250, 3469 ± 330, 3780 ± 310 and 3645 ± 135 ng/mL) on day 7. Specifically, the concentrations were highest (i.e., 4758 ± 301, 4300 ± 210, 4520 ± 400 and 3780 ± 301 ng/mL) on day 9. Levels of secretion were consistently high for 11 d for all clones. These results indicate that the expression level increased depending on the cultivation duration, with the optimal recovery time being 9 d post-cultivation.

### 2.4. Western Blotting for eel FSH-M Detection

Eel FSH-M was isolated from seven clones. The supernatant was collected on day 9 post-cultivation. Twenty microliters of the medium were used for Western blotting. Two specific bands were detected in the range of 39–46 kDa (Figure 4A). The molecular weight of eel FSH-M was approximately 6–8 kDa higher than that of FSH-wt, suggesting that the eCTP β-subunit linker was successfully added between the β-subunit and α-subunit of the tethered eel FSH-wt. Clones 3 and 8 showed stronger signals than the other clones. Then, the secretion pattern was identified according to the culture time in clones 3 and 8, showing that the band intensity increased gradually (Figure 4B). Weak bands were detected on day 5 post-culture. The band intensity was identified as a strong signal on days 7 and 9. The western blot raw date of rec-eel FSH-M were shown at Appendix A.

### 2.5. Deglycosylation the rec-eel FSH-wt and FSH-M Proteins

Western blot analyses of rec-eel FSH-wt and FSH-M revealed an approximate molecular weight of 34–40 kDa and 39–46 kDa, respectively (Figure 2A and Figure 4A). However, PNGase F treatment of the rec-eel FSH-wt decreased the molecular weight to approximately 26–27 kDa. We also compared rec-eel FSH-M with the wild type. The molecular weight of FSH-M was higher (by approximately 6–8 kDa) than that of the wild type, at 39–46 kDa (Figure 5). To further characterize rec-eel FSH-M, we digested rec-eel proteins with PNGase F (Figure 5). Treatment with PNGase F decreased the molecular weight of eel FSH-M protein to 34 kDa, similar to that of eel FSH-wt. The eCTP β-subunit linker region only had 35 amino acids, including 12 potential *O*-linked glycosylation sites, demonstrating that the increased molecular weight corresponded to additional oligosaccharides in *O*-linked glycosylation sites. However, it is difficult to determine the precise molecular weight of carbohydrate using Western blotting owing to the broad bands of attached oligosaccharides. Thus, rec-eel FSH-wt and FSH-M proteins produced from DG 44 CHO cells confirmed that four N-glycosylation sites at Asn^56^ and Asn^79^ of α-subunit as well as Asn^5^ and Asn^22^ of the β-subunit were post-translationally modified. Since we did not analyze the O-glycosidase treatment, the addition of the oligosaccharide chain to the glycosylation sites present in the eCTP β-subunit linker cannot be verified. The deglycosylated western blot raw date of rec-eel FSH-wt and FSH-M was presented to Appendix A.

### 2.6. Secretion of rec-eel FSH-M

To analyze the quantities of secreted protein in the seven eel FSH-M clones, the supernatants were collected on days 1, 3, 5, 7, and 9 post-culture. The quantity of rec-eel FSH-M protein increased gradually (Figure 6). Rec-FSH-M levels were 578–1790 ng/mL on day 3, similar to the results for eel FSH-wt. The concentrations increased rapidly 2167–3945 ng/mL on day 5, reached 3465–4450 ng/mL on day 7, and remained steady until day 9. The highest concentrations were 4450 ± 250 and 4620 ± 120 ng/mL on days 7 and 9 for clone 3, respectively. The concentrations were high on days 7 and 9 post-cultivation for all clones. Thus, tethered rec-eel FSH-M in CHO DG44 cells was collected on day 9 post-cultivation.

### 2.7. Biological Activities of rec-eel FSH-wt and FSH-M

In vitro biological activities were assessed using transfected CHO-K1 cells expressing the FSH receptor, as reported previously [5]. cAMP activation in rec-eel FSH-wt and FSH-M is summarized in Table 1. The dose–response curve for eel FSH-M was slightly left-shifted compared with that of eel FSH-wt (Figure 7). Rec-eel FSH-wt also exhibited full biological activity, with an EC_50_ value of 159.5 ng/mL and Rmax of 62.8 ± 0.8 nM/10^4^ cells. The biological activity of eel FSH-M was higher than that of FSH-wt, with an EC_50_ and Rmax values of 129.2 ng/mL and 67.1 ± 0.7 nM/10^4^ cells, respectively (Table 1). 

The EC_50_ and Rmax values were 1.23- and 1.06-fold higher, respectively, indicating that the FSH-M showed slightly higher biological activity than FSH-wt. This may be attributed to the attachment of the eCG β-subunit CTP linker containing approximately 12 *O*-linked glycosylation sites. Therefore, the eel FSH-M with the eCG β-subunit CTP linker in the middle of tethered eel FSH-wt contributes to signal transduction through those receptors. Therefore, tethered rec-eel FSH-M with the eCG β-subunit CTP linker enabled the large-scale production of stable expression in CHO-DG44 cells. Thus, we established a large-scale production system for rec-eel FSH-wt and FSH-M.

### 2.8. pERK2/1 Activation of the rec-eel FSH-wt and FSH-M Proteins

The total ERK1/2 ratio was 11,407–12,115, as determined by time-dependent or dose-dependent assays. Negative controls had total ERK1/2 and pERK1/2 ratios of 329 and 299, respectively. pERK1/2 ratios after the eel FSH-wt and FSH-M treatment were 2247 ± 73 and 1998 ± 43 at 0 min, respectively. At 5 min, pERK1/2 levels increased sharply to 7457 ± 51 and 7091 ± 79 and then decreased to 3511 ± 225 and 3901 ± 238 at 30 min post-agonist treatment (Figure 8A). The pERK1/2 levels did not differ significantly between eel FSH-wt and FSH-M treatments. Basal pERK1/2 levels were slightly lower in the eel FSH-M-treated group than in the FSH-wt group. Subsequently, the ratios observed at 0 min were set to 1, and the results for various time points are presented as fold change values (Figure 8B). The ratios for eel FSH-wt and FSH-M at 5 min were 3.32- and 3.51-fold higher than those at 0 time, respectively, and decreased to 1.56- and 1.92-fold at 30 min, respectively. We also analyzed pERK1/2 activation by Western blotting (Figure 9A,B). After agonist stimulation, pERK1/2 activity was highest at 5 min and decreased to 20% of the maximal level at 30 min. The reductions in pERK1/2 levels were consistent with the homogeneous time-resolved fluorescence (HTRF) results. In the present study, we suggest that the pERK1/2 pathway contributes to signaling through the eel FSH receptor, as in most GPCRs. The pERK1/2 raw date of rec-eel FSH-wt and FSH-M were presented to Appendix A.

## 3. Discussion

Oligosaccharides in glycoprotein hormones play roles in several physiological processes, including specific signal transduction pathways, such as PKA, PKC, β-arrestin/GRKs, and pERK1/2. Here, we characterized rec-eel FSH-wt and FSH-M as Gαs-biased FSHR agonists that activate cAMP. These proteins also activate pERK1/2 via other signal transduction pathways. In this study, the biological activities of Anguillid eel FSH-wt and FSH-M produced by CHO-DG44 cells were characterized in vitro, revealing that eel FSH-M showed more potent activity than eel FSH-wt. In glycoprotein hormones, the CTP regions in the eCG and hCG β-subunit are essential for early expression and biological activity. 

With respect to levels of secretion, the number of single-cell clones isolated increased gradually until the final supernatant was collected on days 9–11, with the highest concentration approximately 4000–5000 ng/mL. These results are consistent with the Anguillid eel FSH-wt secretion pattern observed in a transient transfection study. However, eel FSH-M showed a distinct transient expression pattern, with the highest secretion from days 1 to 9 post-transfection [22], consistent with previous findings showing that levels of an hFSH β-hCTP-α mutant increased rapidly [39]. The tethered eCG β/α-wt was highly expressed on day 1 post-transfection and maintained until day 9. However, secretion time was delayed by about 2–3 d due to the removal of the eCG β-subunit CTP region [6]. The secretion rate of hCG lacking β-subunit CTP region is 3- to 4-fold more delayed than that of hCG-wt [40]. In studies of dimeric hFSH α/β, levels of 40–45 mIU/mL and 25–28 mIU/mL were reported in CHO-DG44 and CHO/dhfr cells after MTX selection, indicating that dimeric hFSH shows low expression [17]. Thus, the pattern of secretion differs slightly between transient and stable cells, demonstrating that the CG β-subunit CTP region is involved in the early stage of secretion in transiently transfected cells [22]. Consistent with these observations, rec-eel FSH-M was efficiently secreted into the medium of CHO-DG44 cells, providing a system capable of large-scale production. Therefore, the CTP region in the eCG β-subunit is indispensable for secretion in the mammalian cells.

The molecular weight of the eel α-subunit (which is shared among taxa) purified from the eel pituitary was 17–19 kDa, and the FSH β-subunit shows two bands between 17 kDa and 21 kDa [41]. Thus, dimeric FSH α/β from the eel pituitary linked with a non-covalent bond was approximately 34–40 kDa. Recently, we reported that rec-eel FSH-wt is approximately 34–40 kDa; however, the molecular weight of eel FSH-M increased to 42–45 kDa after the attachment of the eCG β-subunit CTP linker in transiently transfected CHO-S cells [22]. In the present study, rec-eel FSH-wt and FSH-M in all isolated clones showed distinct sizes of 34–40 kDa and 39–46 kDa, respectively, despite loading only 20 µL in Western blotting assays. These ranges were broader than those in transiently transfected cells. The increased molecular weight of eel FSH-M was explained by the eCG β-subunit CTP attachment. The increase in molecular weight was approximately 6–8 kDa due to the presence of 35 amino acids only, including 12 *O*-linked glycosylation sites. The oligosaccharides were appropriately modified in CHO-DG44 cells. PNGase F treatment clearly reduced the molecular weights to 26 and 34 kDa, respectively, indicating that 8–10 kDa oligosaccharides were modified. Our results were consistent with those of previous studies, suggesting that hCTP and eCTP linker attachments on hFSH β-hCTP-α [18], hTSH β-hCTP-α [42], hCTP-hGH-hCTP-hCTP [16], eel LH-M, and eel FSH-M [22] increase molecular weights remarkably. 

Glycoprotein hormones, including hCG, eCG, eFSH, Anguillid eel FSH, and Anguillid eel LH, are indicators of the cAMP responses [2,6,43], ovarian superstimulatory response and embryonic development by rec-eCG [44,45], and in vivo potency of rec-hCG [46]. Many studies have reported that the biological activity of rec-glycoprotein hormones is dramatically reduced by deglycosylation at specific glycosylation sites [46,47,48,49]. Specific glycosylation sites at eel LH α-subunit (Asn^56^) and LH β-subunit (Asn^10^) are indispensable in PKA signal transduction in cells expressing eel LH/CGR [7]. Deglycosylated eel FSH mutants show dramatic decreases in EC_50_ values and maximal cAMP responsiveness in vitro [5]. In this study, we also demonstrated that rec-eel FSH-M showed higher PKA/cAMP responsiveness than eel FSH-wt. Our data are consistent with previous results for hFSH [40] and eel FSH [22], indicating that the CTP linker regions of the hCG β-subunit and eCG β-subunit, including *O*-linked glycosylation sites, play a considerable role in determining the biological activity. 

In key studies of hCG CTP linker attachment and the pharmacokinetics of hGH MOD-4023 with copies of the hCG β-subunit CTP linker, the half-life, mean residence time, and time of maximal plasma concentration were dramatically higher than those of a commercial hGH (Biotropin) [16]. A recent study of Japanese and Caucasian adult suggested that MOD-4023 has a favorable safety profile and local tolerance following single-dose subcutaneous administration as a long-acting formulation for once-weekly administration [50]. MOD-4023 in a phase 2 study was found to be clinically effective after once-weekly administration as an alternative to daily injections in adults with a GH deficiency [25,26]. Thus, long-acting formulations of MOD-4023 are expected to obviate the need for repeated injections. Although the in vivo experiment was not conducted in the present study, rec-eel FSH-M could represent a longer-acting formulation. Therefore, in vivo experiments are needed to evaluate this.

The phosphorylation of ERK1/2 involves the sequential activation of three kinases: Raf1, MEK1, and ERK1/2 [51]. The effects of cAMP on pERK1/2 activation are complex and cell-type specific [52]. It has been demonstrated for FSHR that the ERK signaling pathway is involved in the coordinated activation of G protein-dependent and β-arrestin-dependent mechanisms, with distinct temporal and spatial characteristics [32,37,53]. In this study, pERK1/2 showed a peak response after 5 min of stimulation. There was a 3.32- to 3.51-fold increase over the basal response to the agonist. These results are consistent with those for hFSH-stimulated pERK1/2 activation, showing a peak at approximately 6 min and then a slow decline [38]. They also reported that the increase in β-arrestin-mediated FSH-R internalization had no effect on the ERK response in β-arrestin 1- and 2-transfected HEK293 cells. In other studies of hFSHR, peak pERK1/2 activation was observed at 5 min post-stimulation, after which it decreased gradually over time in hFSHR-wt. However, six FSHR-specific residues in extracellular loop 2 (EL2) result in impaired internalization and low cAMP responsiveness [54]. The hFSHR EL3M mutant appears to be important for internalization and pERK1/2 activation [55]. EL2 and EL3 mutants hamper β-arrestin recruitment to the activated receptors, hence their internalization in a particular location. Thus, our results are consistent with cAMP responsiveness and pERK1/2 activation through the FSH-FSHR complex in response to the two agonists. Overall, these results demonstrate that the FSHR-mediated signaling pathway is followed by pERK1/2 activation. The PKA signaling pathway is activated by the cellular cAMP response, and pERK1/2 further phosphorylates downstream effectors of the MAPK pathway. However, we cannot conclude whether pERK1/2 activation is transmitted down in a Gαs-biased or β-arrestin-biased manner. 

## 4. Materials and Methods

### 4.1. Materials

Oligonucleotides were synthesized by Genotech (Daejeon, Republic of Korea). The pGEMT-easy cloning vector was purchased from Promega (Madison, WI, USA). The Lipofectamine-2000 and Freestyle MAX reagent were purchased from Invitrogen (Carlsbad, CA, USA). The pOptiVEC^TM^-TOPO TA Cloning Kit, Freedom^TM^ DG44 Kit, DG44 CHO cells, CD DG44 medium, CD OptiCHO^TM^ medium, CD FortiCHO^TM^ medium, methotrexate (MTX) reagent, and cloning media were purchased from Life Technologies (Carlsbad, CA, USA). CHO-K1 cells and HEK 293 cells were obtained from the Korean Cell Line Bank (Seoul, Republic of Korea). The pCORON1000 SP VSV-G tag expression vector was purchased from Amersham Biosciences (Piscataway, NJ, USA). Fetal bovine serum (FBS), Ham’s F-12 medium, OptiMEM, and CHO-S-SFMII medium were purchased from Gibco BRL (Grand Island, NY, USA). A monoclonal antibody (designated as 5A14) for Western blotting was produced, and a rec-eel FSH ELISA system (designated as 5A11 and 5A14) was developed in our laboratory [5]. DNA ligation reagents, endonucleases, polymerase chain reaction (PCR) reagents, and restriction enzymes were purchased from Takara Bio (Shiga, Japan). A deglycosylation Kit (PNGase F), used to remove *N*-linked glycosylation, was purchased from New England Biolabs (Ipswich, MA, USA). The cAMP Dynamic 2 Immunoassay Kit and pERK1/2 Kit were purchased from Cisbio (Codolet, France). The QIAprep-Spin Plasmid Kit was purchased from Qiagen Inc. (Hilden, Germany), and disposable spinner flasks were purchased from Corning Inc. (Corning, NY, USA). Centriplus Centrifugal Filter Devices were purchased from Amicon Bio Separation (Billerica, MA, USA). All other reagents were purchased from Sigma-Aldrich (St. Louis, MO, USA). 

### 4.2. Construction of eel FSH-wt and FSH-M Vectors 

The cDNAs of eel FSH α- and β-subunit were cloned using the cDNA from Japanese eel *Anguilla japonica* pituitary, as previously reported [7]. We constructed single-chain eel FSH-wt-linked α-subunit without the signal sequence at the C-terminal region of FSH β-subunit. Finally, an FSH mutant, FSH-M, was also constructed with the eCG β-subunit C-terminal region (eCTP: 115–149 amino acids) attached between the β-subunit and α-subunit by PCR, as reported previously [22]. The full-length PCR products were ligated into the pGEMT-Easy vector. The clones were used to transform DH5α-competent cells. Plasmids were isolated and sequenced for genetic verification. Full fragments of eel FSH-wt and FSH-M mutant were ligated into the pOptiVEC TOPO TA Cloning expression vector. Finally, the direction of insertion was confirmed by restriction enzyme cutting. Figure 10 presents a schematic representation of eel FSH-wt and eel FSH-M (in which eCTP was attached between the β-subunit and α-subunit). The plasmids were purified and sequenced in both directions by automated DNA sequencing to ensure that the correct mutations had been introduced (designated as pOptiVEC-eel FSHβ/α-wt and FSHβ/α-M).

### 4.3. Transfection into CHO DG44 Cells 

For rec-protein production, the expression vectors were linearized by cutting the Pvu 1 restriction enzyme and transfected into CHO DG44 cells using FreeStyle^TM^ MAX reagent according to the supplier’s protocol. CHO DG44 cells were incubated in complete CD DG44 medium at 3 × 10^5^ viable cells/mL on an orbital shaker platform rotating at 130–135 rpm at 37 °C and 8% CO_2_. One day prior to transfection, CHO DG44 cells were passaged at a density of 3 × 10^5^ cells/mL. On the day of transfection, the cell density was approximately 5 × 10^5^ cells/mL. Cell viability was maintained at >95% to ensure optimal transfection. For each transfection, CHO DG44 cells (1.5 × 10^7^ viable cells) were transferred into a new 125 mL spinner flask and pre-warmed; complete CD DG44 medium was added to a final volume of 30 mL. The plasmid DNA (18 µg) was diluted in 600 µL of OptiPRO^TM^ serum-free medium (SFM) and 15 µL of FreeStyle^TM^ MAX Reagent was added to 600 µL of OptiPRO^TM^ SFM. The diluted FreeStyle^TM^ MAX Reagent solution was added to the diluted DNA solution, mixed gently by inversion, and incubated for 10 min at room temperature to allow complexes to form. The DNA-FreeStyle^TM^ MAX reagent complex was added dropwise to the cells while slowly swirling the flask.

### 4.4. Isolation of Single Cells Expressing rec-eel FSH-wt and FSH-M Proteins

At 48 h after transfection, the cells were collected by centrifugation at 300× *g* for 5 min, and the medium was removed by aspiration. The cells were passed through complete CD OptiCHO^TM^ medium (fresh growth medium) supplemented with 8 mM l-glutamine to obtain a final density of 5 × 10^5^ viable cells/mL for selection. Fresh growth medium was added to a 30 mL volume every 3–4 d for approximately 10–14 d until cell viability increased to over 90%. Next, the cells were adapted to amplify the integration locus of the plasmid and potentially increase protein production using MTX reagent. First, we amplified the cells at 500 nM MTX for approximately 3 weeks until cell viability reached over 95%. Then, additional rounds of MTX amplification were increased to 2 µM for approximately 14 d and 4 µM for 14 d. Finally, the cell survival rate was >95%. The selected cell pools by MTX amplification were frozen in aliquots at −80 °C.

Next 3 × 10^5^ viable cells/mL were seeded in a sufficient culture volume of fresh growth medium supplemented with 8 mM l-glutamine to generate a sufficient conditioned medium for cloning experiments. The cells were grown in batch culture for 5 d. The cultures were then centrifuged, and the supernatant was collected. It was sterilized by membrane filtration and frozen in aliquots at −80 °C until needed. To limit dilution cloning, the MTX-amplified cells were expanded in fresh growth medium without 8 mM glutamine for at least two passages. On the day of cloning, the cells were serially diluted to a final concentration of 1000 cells/mL in fresh growth medium supplemented with 8 mM glutamine. Then, 0.2 mL of the cell suspension was diluted with 100 mL of complete cloning medium (86 mL of basal CD FortiCHO^TM^ medium, 3 mL of freshly thawed 200 mM l-glutamine, 10 mL of conditioned medium, and 1 mL of 100 × HT supplement). The diluted cells were dispensed into 0.5–2 cells per well in a 96-well plate. After 14 d of incubation, the wells were visually examined under a microscope to evaluate the growth of monoclonal colonies. After isolating the clones, the single-cell colonies were transferred to 24-well plates in 0.5–1 mL of fresh growth medium supplemented with 6 mM l-glutamine. The culture volumes in six-well plates and T-25 flasks were 2–3 mL and 5–7 mL, respectively. Finally, the clones were expanded in 125 mL shaker flasks and the cells were incubated at 37 °C and 8% CO_2_ with shaking at 130–135 rpm.

### 4.5. Production of rec-eel FSH-wt and FSH-M Proteins

The single clone cells were incubated at 37 °C in a humidified atmosphere of 8% CO_2_ on an orbital shaker platform rotating at 135 rpm. To check for protein production, single clone cells were seeded at 3 × 10^5^ viable cells/mL in 30 mL of fresh medium supplemented with 4 mM l-glutamine. Then, 2 mL of culture medium was recovered on days 0, 1, 3, 5, 7, 9, and 11 until the cell viability dropped below 50%. The collected samples were analyzed for the secreted rec-protein. Finally, the culture media were collected on days 9 and 11 post-seeding and centrifuged at 100,000× *g* for 10 min at 4 °C to remove cell debris. The supernatant was collected, and a portion was concentrated using a Centricon filter to adjust for cAMP assay. 

### 4.6. Enzyme-Linked Immunosorbent Assay (ELISA) for rec-eel FSH Protein Quantitation

The rec-eel FSH-wt and FSH-M proteins in cell culture media were quantified by a double-sandwich ELISA using plates coated with the monoclonal antibody eel FSH 5A14 (binds β-subunit of eel FSH) as previously reported [7]. Tethered eel FSH β/α cDNA was constructed and expressed in *Escherichia coli*. Rec-protein was purified by using Ni-NTA Sepharose column chromatography. Mice were immunized with 50 ug of the purified protein and the hybridoma cells were isolated by fusing spleen cells with mouse myeloma Sp2/0 cells. Monoclonal antibodies were purified from culture supernatants by affinity chromatography using a Hi-Trap Protein G column. 

Briefly, the wells were blocked with 1% skim milk in phosphate-buffered saline (PBS) for 1 h and then washed with PBS containing 0.05% Tween 20 (PBS-T). Next, 100 μL of collected media was added and incubated for 1–2 h at 37 °C. After washing with PBS-T three times, 400-fold-diluted horseradish peroxidase (HRP)-conjugated anti-eel FSH 5A11 (binds α-subunit of eel FSH) antibody was added and incubated for 1 h. The wells were washed 5 times and incubated with 100 μL of substrate solution (tetramethylbenzidine) for 20 min. The reaction was stopped by adding 50 μL of 1 M H_2_SO_4_. The absorbance of each well was measured at 450 nm using a microplate reader (Cytation 3, Biotek, Winooski, VT, USA). 

### 4.7. Western Blotting Analysis of rec-eel FSH Proteins

For the Western blotting analysis, the collected sample (20 µL) from the supernatant media was reduced by 12% sodium dodecyl sulfate poly-acrylamide gel electrophoresis (SDS-PAGE). After SDS-PAGE, the proteins were transferred onto a polyvinylidene difluoride (PVDF) membrane (0.2 μm) using a Bio-Rad Mini Trans-Blot electrophoresis cell (Hercules, CA, USA). The membrane was blocked by incubation in 5% skim milk in TBS-T (20 mM Tris-HCl, pH 7.6, 140 mM NaCl, 0.1% Tween 20) and then incubated with a monoclonal anti-eel FSH β antibody (designated as 5A14) for 13–15 h. The blot was washed with TBS-T and incubated with HRP-conjugated anti-mouse secondary antibody for 2 h. Thereafter, the membrane was incubated with 2 mL of Lumi-Light substrate solution for 5 min, and detection was performed using the an enhanced chemiluminescence system.

### 4.8. Enzymatic Release of N-Linked Oligosaccharides

Removal of *N*-Deglycosylation was performed under denaturing conditions according to the manufacturer’s protocol. The rec-eel FSH-wt and FSH-M proteins were analyzed to remove the glycans modified by *N*-glycosylation enzyme treatment. To remove all *N*-linked glycans, rec-protein (15 µg) was incubated for 1 h at 37 °C with PNGase F [1 µL of enzymes (2.5 U/mL)/20 µL sample + 2 µL of 10× Glycobuffer + 2 µL of 10% NP-40] after boiling at 100 °C for 10 min with 1 µL of 10× Glycoprotein Denaturing Buffer. The samples were separated by SDS-PAGE following Western blotting.

### 4.9. Analysis of cAMP Levels by Homogenous Time-Resolved Fluorescence Assays

Transfected cells were subjected to a cAMP analysis 48–72 h post-transfection. cAMP accumulation in CHO-K1 cells expressing eel FSHR was measured using a cAMP Dynamics 2 Competitive Immunoassay Kits. Transfected cells were seeded into 384-well plates at a density of 10,000 cells/well. Compound medium buffer containing the ligand (5 μL) was added to each well and incubated for 30 min. The cAMP response assay used a cryptate-conjugated anti-cAMP monoclonal antibody and a d2-labeled cAMP reagent. Subsequently, cAMP-d2 (5 μL) and anti-cAMP-cryptate (5 μL) were added to each well and incubated for 1 h at room temperature. Compatible homogeneous time-resolved fluorescence (HTRF) energy transfer (665 nm/620 nm) was measured using a TriStar2 S LB942 microplate reader (BERTHOLD Tech, Wildbad, Germany). The results are expressed as Delta F% (cAMP inhibition), calculated as [Delta F% = (standard or sample ratio-sample negative) × 100/ratio negative]. The cAMP concentration for Delta F% values was calculated using GraphPad Prism (version 6.0; GraphPad Software Inc., La Jolla, CA, USA). 

### 4.10. Measurement of pERK1/2 Levels by Homogenous Time-Resolved Fluorescence Assays

The transfected cells expressing eel FSH receptor were plated to 1.5 × 10^4^ cells/8 µL in a 384-well plate with HBSS medium. Rec-eel FSH-wt and FSH-M (500 ng/mL) were added to the cells from 0 to 30 min. Lysis buffer (4 µL) was added at room temperature with shaking. Next, the premixed antibody solutions with the d2 acceptor and Eu^3^+-Cryptate donor-labelled antibodies were added and covered with a plate sealer. The plates were then incubated for 2 h at room temperature. We also detected total-ERK1/2 using a kit reagent to monitor steady-state protein levels in cells. Finally, the plate was read at two different wavelengths (665 nm/620 nm) using a TriStar2 S LB942 microplate reader. The results are represented as a ratio, calculated as [Ratio%= (signal 665 nm/signal 620 nm) × 10^4^]. Ligand-stimulated HTRF ratios were normalized for each experiment as the fold change in the HTRF ratio from unstimulated cells. 

### 4.11. Analysis of Phospho-ERK1/2 Activation by Western Blot

HEK293 cells were seeded into six-well plates and transfected with the pCORON1000 SP VSV-G vector with eel FSHR. pERK1/2 assay was performed 48 h post-transfection. Cells were starved for at least 4 h in serum-free medium and incubated for various durations. The cells were then lysed with RIPA buffer (Sigma-Aldrich) containing a mixture of protease and protease inhibitor cocktail. Equal amounts (20–40 µg) of cellular extracts were separated by 10% SDS-PAGE and transferred to nitrocellulose membranes. pERK1/2 and total ERK1/2 were detected by immunoblotting using rabbit polyclonal anti-phospho-p44/42 MAPK (1:2000) and anti-MAPK1/2 (1:3000), respectively. The membranes were then incubated with HRP-conjugated anti-rabbit and anti-mouse secondary antibodies. Chemiluminescence detection was performed using SuperSignal^TM^ West Pico reagent, and pERK1/2 immunoblots were quantified by densitometry using an Image-Lab (Bio-Rad). 

### 4.12. Data Analysis

Dose–response curves were generated from experiments performed in duplicate. GraphPad Prism 6.0 (San Diego, CA, USA) was used to analyze the cAMP response, EC_50_ values, and stimulation curves. The curves fitted for a single experiment were normalized to background signals measured in mock-transfected cells. The pERK1/2 values for the percentage ratio and Western blot were calculated using GraFit Version 5 (Erithacus Software, Horley Surrey, UK). The results are expressed as the mean ± standard error of the mean of three independent experiments.

## 5. Conclusions

A production system of rec-eel FSH-wt and FSH-M from CHO DG44 cells was successfully established, and the biological activities of rec-eel FSH-wt and FSH-M were evaluated. Rec-eel FSH proteins were generated in large quantities to facilitate artificial ovulation in female eels. The eCTP β-subunit linker for eel FSH improved activity in vitro, and linker attachment was required for maximal cAMP responsiveness and activation of the pERK1/2 signal transduction pathway. Therefore, the rec-eel FSH-M protein is valuable for inducing eel female maturation in vivo. The pERK1/2 activation stimulated by eel FSHR may be mediated by GRKs and β-arrestin isoforms. However, Gαs-biased cAMP signaling needs to be further elucidated to determine its relation to pERK1/2 activation. Therefore, this study identifies the most significant factors on pERK1/2 activation, acting downstream of the cAMP/PKA signal transduction pathway. The eCG β-subunit CTP linker in the middle site of tethered eel FSH-wt plays a pivotal role in signal transduction through these receptors. These findings are extremely important for producing new potent analogs in large quantities using a stable CHO DG44 cell system. The rec-eel FSH-wt and eel FSH-M proteins are promising for large-scale production using CHO-DG44 cells.

## Figures and Tables

**Figure 1 ijms-25-07282-f001:**
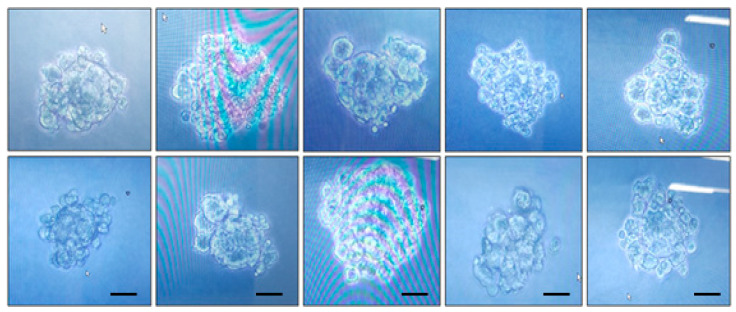
Shape of the colony before isolation from the 96-well plate. After 10 d of incubation, the colonies were visually examined under a microscope for monoclonal colony growth. Images of representative colonies were obtained from eel FSH-wt samples approximately 3 weeks post-plating with complete cloning medium in a non-shaking incubator. Colonies were selected and transferred into 24-well plates. Next, they were transferred into 6-well, T-25 flasks and 125 mL shaker flasks. The scale bars represent 50 μm.

**Figure 2 ijms-25-07282-f002:**
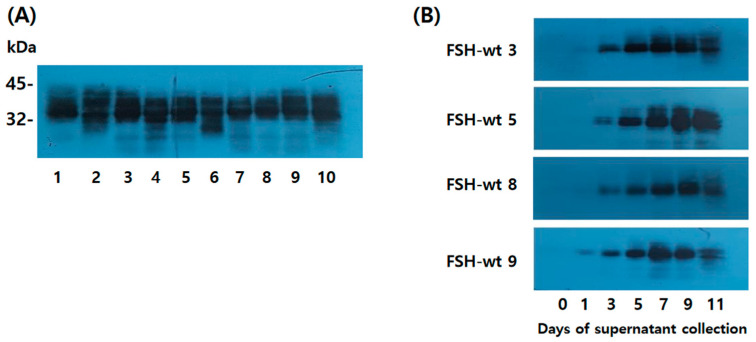
Western blotting analysis of rec-eel FSH-wt proteins produced from single cells. Supernatants from 10 colonies were collected on the day of cultivation in a shaking incubator. The supernatants were resolved by sodium dodecyl sulfate-polyacrylamide gel electrophoresis and blotted onto membranes. Proteins were detected using a monoclonal antibody (anti-eel FSH5A14) and horseradish peroxidase-conjugated goat anti-mouse IgG antibodies. (**A**) In total, 20 µL of the supernatant on day 9 was loaded in the wells. Numbers denote isolated clone counts. (**B**) Colonies with substantial secretion were selected for Western blot analyses. We choose four colonies (eel FSH-wt 3, FSH-wt 5, FSH-wt 8, and FSH-wt 9) and 20 µL of supernatant was evaluated by Western blotting on the day of culture. FSH, follicle-stimulating hormone.

**Figure 3 ijms-25-07282-f003:**
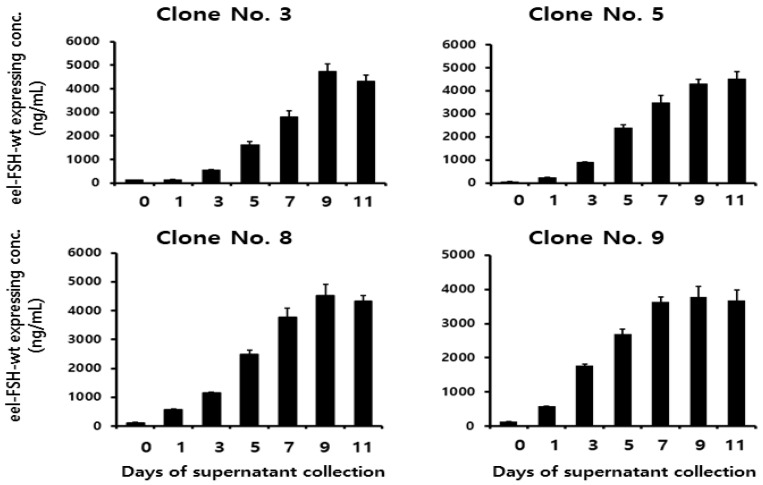
Concentrations of rec-eel FSH-wt proteins secreted from CHO-DG44 cells on the day of culture. The supernatant was collected on days 0, 1, 3, 5, 9, and 11 of culture. The expression levels of rec-eel FSH-wt protein in monoclonal cells were analyzed using a sandwich enzyme-linked immunosorbent assay. Values are expressed as the mean ± standard error of mean from at least three independent experiments. FSH, follicle-stimulating hormone.

**Figure 4 ijms-25-07282-f004:**
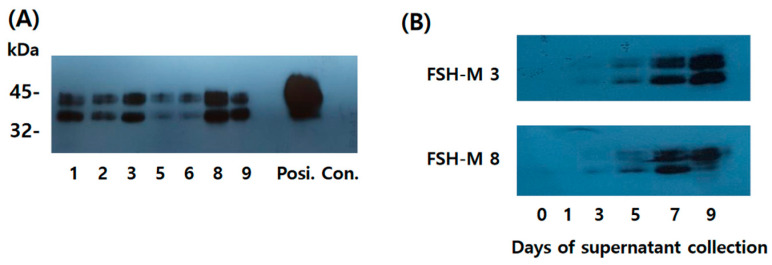
Western blotting analysis of rec-eel FSH-M proteins produced by monoclonal cells. Supernatants were collected from seven colonies on the day of cultivation. The samples were prepared for SDS-PAGE. Membranes were detected using specific monoclonal antibodies (anti-eel FSH5A14). (**A**) In total, 20 µL collected on day 9 was loaded in the wells. Positive controls produced from the CHO-S cells were concentrated by 20 times and 20 µg was loaded in the wells. Two specific bands were detected for all samples. (**B**) Colonies judged to secrete large amounts were selected for Western blot analyses. We choose two colonies (eel FSH-M 3 and FSH-M 8) and 20 µL of the supernatant was used for Western blotting on the day of culture. Faint bands were first detected on day 3 and band intensity increased gradually, reaching a maximum on day 9.

**Figure 5 ijms-25-07282-f005:**
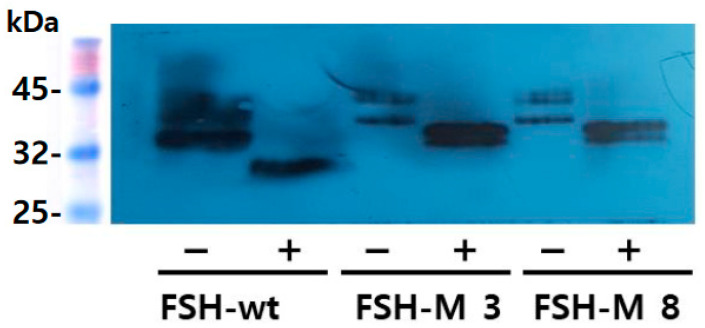
Deglycosylation results for eel FSH-wt and FSH-M proteins. The proteins collected from eel FSH-wt 3, FSH-M 3, and FSH-M 8 were treated with peptide-N-glycanase F to remove *N*-linked oligosaccharides, followed by Western blotting. The molecular weights of rec-eel FSH-wt and FSH-M decreased significantly to approximately 8–10 kDa.

**Figure 6 ijms-25-07282-f006:**
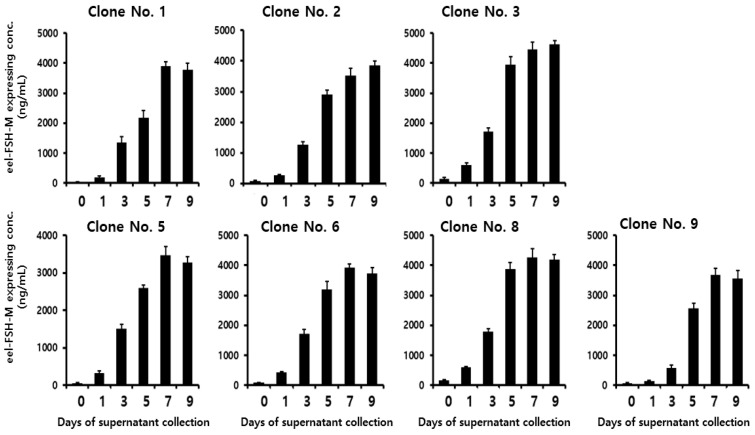
Concentrations of rec-eel FSH-M proteins secreted from CHO-DG44 cells on the day of culture. The supernatants were collected on days 0, 1, 3, 5, 9, and 11 of culture. The expression levels of rec-eel FSH-M in monoclonal cells were analyzed using a sandwich enzyme-linked immunosorbent assay. Values are expressed as the mean ± standard error of mean from at least three independent experiments.

**Figure 7 ijms-25-07282-f007:**
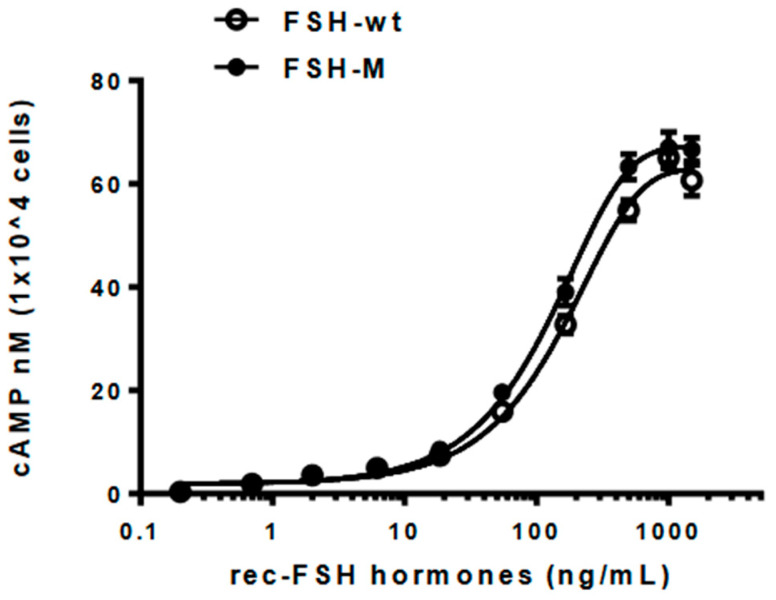
Effects of rec-eel FSH-wt and FSH-M on cyclic adenine monophosphate (cAMP) production in cells expressing eel follicle-stimulating hormone receptor. CHO-K1 cells transiently transfected with eel FSHR were seeded in 384-well plates (10,000 cells per well) at 24 h post-transfection. Cells were incubated with rec-eel FSH-wt or FSH-M for 30 min at room temperature. cAMP production was detected using a homogeneous time-resolved fluorescence assay and results are represented as Delta F%. Each data point represents the mean ± standard error of mean from triplicate experiments. The mean values were fitted to an equation to obtain a single-phase exponential decay curve.

**Figure 8 ijms-25-07282-f008:**
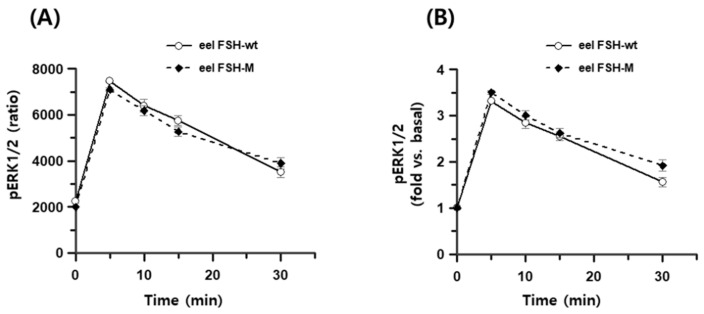
Time course for pERK1/2 activation in rec-eel FSH-wt and FSH-M. HEK293 cells were transiently transfected with eel FSH receptor, stimulated with 400 ng/mL agonist and normalized to the basal response. (**A**) Ratios are shown as delta F%. (**B**) The folds change values are shown, with 0 time set to 1-fold.

**Figure 9 ijms-25-07282-f009:**
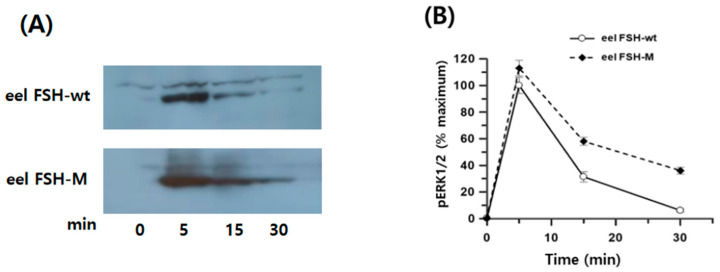
pERK1/2 activation stimulated by eel FSH receptor. The eel FSH receptor was transiently transfected into HEK293 cells, and the cells were starved for 4–6 h and stimulated with a 400 ng/mL agonist for the indicated times. Whole-cell lysates were analyzed for pERK1/2 and total ERK levels. Twenty micrograms of protein were used in each sample lane. (**A**) Phosphorylation of ERK1/2 by western blotting. (**B**) The pERK and total ERK bands were quantified by densitometry, and pERK was normalized to total ERK levels. Representative data are shown, and graphs represent the mean and SE values from three independent experiments. No significant differences were observed between the curves representing eel FSH-wt- and FSH-M-treated samples.

**Figure 10 ijms-25-07282-f010:**
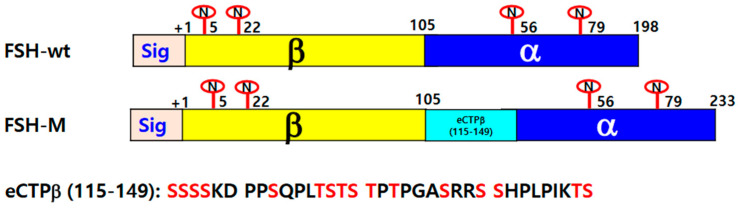
Schematic diagram of rec-eel FSH-wt and eel FSH-M. The tethered form of eel FSH β/α-wt containing the β-subunit and common α-subunit sequences was engineered. In the eel FSH-M mutant, the eCG β-subunit carboxyl-terminal peptide linker was inserted between the β-subunit and α-subunit using polymerase chain reaction. The eCG β-subunit CTP linker contained 35 amino acids sequence corresponding to the carboxyl-terminal peptide of the eCG β-subunit with approximately 12 *O*-linked oligosaccharide sites. The numbers in FSH-wt and FSH-M indicate the amino acids of the mature protein, except for the signal sequences. “N” denotes *N*-linked glycosylation sites at the eel FSH β-subunit and FSH α-subunit. Yellow indicates FSH β, the α-subunit is shown in blue, and the light blue shows the eCG CTP linker. eCTPβ (115–149) is the amino acid sequences of the eCG β-subunit CTP linker. Red in eCTPβ (115–149) denotes potential *O*-linked oligosaccharide sites.

**Table 1 ijms-25-07282-t001:** Bioactivity of rec-eel FSH proteins in cells expressing eel FSH receptor.

rec-FSH Hormones	cAMP Responses		
	Basal *^a^* (nM/10^4^ cells)	Log (EC_50_) (ng/mL)	Rmax *^b^* (nM/10^4^ Cells)
FSH-wt	1.9 ± 0.5	159.5 (144.0 to 178.7) *^c^*	62.8 ± 0.8
FSH-M	1.8 ± 0.4	129.2 (118.9 to 141.6)	67.1 ± 0.7

Values are the means ± SEM of triplicate experiments. Log (EC_50_) values were determined from the concentration-response curves from in vitro bioassays. *^a^* Basal cAMP level average without agonist treatment. *^b^* Rmax average cAMP level/10^4^ cells. *^c^* Geometric mean (95% confidence intervals) of at least three experiments.

## Data Availability

The original contributions presented in the study are included in the article/Appendix A, further inquires can be directed to the corresponding author.

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
