# Peer review of "Stable Production of a Recombinant Single-Chain Eel Follicle-Stimulating Hormone Analog in CHO DG44 Cells"

_ijms, 2024, doi:10.3390/ijms25137282_

Round 1

Reviewer 1 Report

Comments and Suggestions for Authors

This article focuses on expression and stable production of a recombinant single chain Eel Follicle Stimulating Hormone analogs in CHO DG44 cell line. The authors have generated a mutant version of this protein where they have introduced an eCG β-subunit CTP fragment between the β-subunit and α-subunit of the wild type single chain eel FSH (FSH-WT). The authors have verified the expression of both the WT and the mutant FSH over a period and shown the optimal duration for the maximal expression of each protein. The authors also tested the activity of both the proteins and showed that the mutant FSH has slightly higher biological activity than the WT. The authors have done a decent work of characterizing the protein and has shown the importance of eCTP β-subunit in enhancing the yield and activity of the eel FSH protein. However, I have some questions and concerns regarding some of the results. I think there are some instances where authors could have provided more information and could have included some more work to make this work stronger and more suitable for this journal.

I think the authors need to explain or include more information about the importance of why high production of eel FSH is essential. This article does not provide the overall goal of this work. The authors should include a paragraph in introduction, about the need for and importance of eel FSH.

The English needs to be improved. There are multiple instances where the sentences are difficult to understand. 

Also, are the WT and mutant FSH the only proteins that are released in the media? The authors should have included a Coomassie gel to show the total protein content in the media.

Have the authors tried to purify the specific protein from the media and characterize it? If this protein is going to be produced at high levels for pharmaceutical purposes, it is a good idea to purify and characterize these proteins in vitro. Long term stability of these proteins needs to be characterized too.

Line 41: The α-subunit is composed of two carbohydrate adhesive sites at Asn56 and Asn79. I would just use the term glycosylation site to maintain the uniformity.

I would move section 2.6 in the materials and methods section right after section 2.4 to maintain the continuity.

Line 105: The figure numbers are mislabeled in the paper. It says figure 1, but it is figure 2 based on the attached figures in the paper.

Line 70: The authors mention about the hGH-MOD-4023 but fail to provide more information about it or any reference regarding it.

Figure 3A/B: FSH-WT western blots. Have the authors looked at the expression level beyond 11 days? It seems clone 5 was still going up at 11 days.

Results section 2.6: Authors mention that they collected supernatants up to 11 days, but the figure only has data up to 9 days. Again, for clone 2 and 3, the FSH-M level is still going up. Did author verify the expression beyond 9 or 11 days?

Line 327: Did the author mean figure 1 instead of figure 10?

Comments on the Quality of English Language

The English needs to be improved. There are multiple instances where the sentences are difficult to understand. 

Line 105: Needs to be restructured to make it understandable.

Line 153: "glycosylation attachment post-translational" This statement needs to be re written. 

There are more instances, where it is hard to understand the sentence. Mostly the sentences need to be reframed.

Author Response

Reviewer 1

Comments and Suggestions for Authors

This article focuses on expression and stable production of a recombinant single chain Eel Follicle Stimulating Hormone analogs in CHO DG44 cell line. The authors have generated a mutant version of this protein where they have introduced an eCG β-subunit CTP fragment between the β-subunit and α-subunit of the wild type single chain eel FSH (FSH-WT). The authors have verified the expression of both the WT and the mutant FSH over a period and shown the optimal duration for the maximal expression of each protein. The authors also tested the activity of both the proteins and showed that the mutant FSH has slightly higher biological activity than the WT. The authors have done a decent work of characterizing the protein and has shown the importance of eCTP β-subunit in enhancing the yield and activity of the eel FSH protein. However, I have some questions and concerns regarding some of the results. I think there are some instances where authors could have provided more information and could have included some more work to make this work stronger and more suitable for this journal.

I think the authors need to explain or include more information about the importance of why high production of eel FSH is essential. This article does not provide the overall goal of this work. The authors should include a paragraph in introduction, about the need for and importance of eel FSH.

→We inserted the goal of this work as reviewer’s comment.

The English needs to be improved. There are multiple instances where the sentences are difficult to understand.

→We re-edited by special English editing company as reviewer’s comment.

Also, are the WT and mutant FSH the only proteins that are released in the media? The authors should have included a Coomassie gel to show the total protein content in the media.

→We inserted “Coomassie gel result” in the supplementary Figure 1.

Have the authors tried to purify the specific protein from the media and characterize it? If this protein is going to be produced at high levels for pharmaceutical purposes, it is a good idea to purify and characterize these proteins in vitro. Long term stability of these proteins needs to be characterized too.

→We are currently building a refining system. Thus, we are going to apply with a specific antibody.

Line 41: The α-subunit is composed of two carbohydrate adhesive sites at Asn56 and Asn79. I would just use the term glycosylation site to maintain the uniformity.

→We changed “carbohydrate adhesive sites” to “glycosylation sites” by reviewer’s comment

I would move section 2.6 in the materials and methods section right after section 2.4 to maintain the continuity.

→I don’t understand exactly if it’s is Results or Materials and methods section.

Line 105: The figure numbers are mislabeled in the paper. It says figure 1, but it is figure 2 based on the attached figures in the paper.

→We mispresented the figure order. Thus, we rearranged figure order.

Line 70: The authors mention about the hGH-MOD-4023 but fail to provide more information about it or any reference regarding it.

→We inserted “This substance was finally reported to be successful as a result of a safety and dose-finding study in clinical trials for growth hormone deficient (GHD) children [26].” after the Line 70.

Figure 3A/B: FSH-WT western blots. Have the authors looked at the expression level beyond 11 days? It seems clone 5 was still going up at 11 days.

→We did not collect the supernatant after 11 days post-cultivation. At the 11 days, the all supernatants were collected. Thus, we could not check the expressing quantity of rec-FSH-wt

Results section 2.6: Authors mention that they collected supernatants up to 11 days, but the figure only has data up to 9 days. Again, for clone 2 and 3, the FSH-M level is still going up. Did author verify the expression beyond 9 or 11 days?

→We changed “1, 3, 5, 7, 9, and 11 post-cultivation” to “1, 3, 5, 7, and 9 post-cultivation in the Line 163.

ve did the isolation of FSH-wt clone cells first. And we decide to collect the supernatants until 9 days with reference to FSH-wt analysis results and the results of the other DG 44 cells.

Line 327: Did the author mean figure 1 instead of figure 10?

→ All Figures were rearranged

Comments on the Quality of English Language

The English needs to be improved. There are multiple instances where the sentences are difficult to understand. 

→We re-edited by special English editing company as reviewer’s comment.

Line 105: Needs to be restructured to make it understandable.

→The sentence was changed to “Finally, 10 monoclonal cells expressing rec-eel FSH-wt were isolated and stored in a LN2 tank. Next, an experiment was conducted to confirm the production of rec-FSH proteins by culture day.”.

Line 153: "glycosylation attachment post-translational" This statement needs to be re written. 

There are more instances, where it is hard to understand the sentence. Mostly the sentences need to be reframed.

The sentences were rewritten “Thus, rec-eel FSH-wt and FSH-M proteins produced from DG 44 CHO cells confirmed that four N-glycosylation sites at Asn56 and Asn79 of α-subunit and at Asn5 and Asn22 of the β-subunit were post-translational modification. Since we did not analyze the experiment of O-glycosidase treatment, we cannot be sure of oligosaccharide chain addition for the glycosylation sites present in the eCTP β-subunit linker.

Reviewer 2 Report

Comments and Suggestions for Authors

The manuscript with ID (ijms-3027470) by Byambaragchaa and coauthors studied the potential production of a recombinant single-chain eel (rec-eel FSH molecule Analog) in CHO DG44 cells. The study is interesting and within the scope of the journal. I have 3 questions only: -

Q1. Line 306: How did the authors produce a monoclonal antibody in the laboratory? Details should be described for possible replication of your findings.

Q2. Lines 318-319: From which eel species the authors sampled the cDNAs of eel FSH α- and β-subunits?

Q3. I found that presentation of the supplementary files of gels in the main manuscript text is important. 

Comments on the Quality of English Language

Moderate editing of English language required

Author Response

Comments and Suggestions for Authors

The manuscript with ID (ijms-3027470) by Byambaragchaa and coauthors studied the potential production of a recombinant single-chain eel (rec-eel FSH molecule Analog) in CHO DG44 cells. The study is interesting and within the scope of the journal.

I have 3 questions only: -

Q1. Line 306: How did the authors produce a monoclonal antibody in the laboratory? Details should be described for possible replication of your findings.

We inserted “Tethered eel FSHb/a cDNA was constructed and expressed in E.coli. Rec-protein was purified by using Ni-NTA Sepharose column chromatography. Mice were immunized with 50 ug of the purified protein and the hybridoma cells were isolated by fusing spleen cells with mouse myeloma Sp2/0 cells. Monoclonal antibodies were purified from culture supernatants by affinity chromatography using a Hi-Trap Protein G column.” For the antibody production In the Section 4.6.

Q2. Lines 318-319: From which eel species the authors sampled the cDNAs of eel FSH α- and β-subunits?

 We inserted “from Japanese eel Anguilla japonica pituitary” in the Line 329.

Q3. I found that presentation of the supplementary files of gels in the main manuscript text is important. 

Gel files in the main manuscript text were inserted as the supplementary files.

Comments on the Quality of English Language

Moderate editing of English language required

All manuscripts were reedited by special English editing company.

Reviewer 3 Report

Comments and Suggestions for Authors

IJMS 3027470

 Article 

Stable Production of a Recombinant Single-Chain Eel Follicle-Stimulating Hormone Analog in CHO DG44 Cells 

 Munkhzaya Byambaragchaa 1,2, Sei Hyen Park 3, Sang-Gwon Kim 3, Min Gyu Shin 4, Shin-Kwon Kim 4, Myung-Hum Park 5, Myung-Hwa Kang and Kwan-Sik Min 1,2,7,

  Abstract: The abstract is well presented. Chapter does not require vital additions or changes. For a better understanding authors should give scientific names of eel and Chinese hamster (rows 15-16).   Keywords expresses the content of the work.   Chapter 1. Introduction. The introduction chapter presents the importance of hormones, a description of them as well as results on recombinant hormone in Chinese hamster ovary cells. Authors should give scientific names of Chinese hamster (row 50), eel (row 64)   Chapter 2. Materials and Methods.    The Materials and methods chapter is composed of twelve subchapters as follows: (new numbered)

2.1. Materials 

2.2. Construction of eel FSH-wt and FSH-M vectors 

2.3. Transfection into CHO DG44 cell 

2.4. Isolation of single cells expressing rec-eel FSH-wt and FSH-M proteins 

2.5. Production of rec-eel FSH-wt and FSH-M proteins 

2.6. ELISA for rec-eel FSH protein quantitation 

2.7. Western blotting analysis of rec-eel FSH proteins 

2.8. Enzymatic release of N-linked oligosaccharides 

2.9. Analysis of cAMP levels by homogenous time-resolved fluorescence assays 

2.10. Measurement of pERK1/2 levels by homogenous time-resolved fluorescence assays 

2.11. Analysis of phospho-ERK1/2 activation by western blot 

2.12. Data Analysis 

Authors should change in paper the order of chapters Chapter 4 to become Chapter 2 and vice versa, and also the 12 sub-chapters to be summarized in such a way as to present clearly the materials and methods used.   Chapter 3. Discussion. Chapter 3 Discussions is well-proportioned and makes the necessary and informative references on the research results. Chapter 3. does not require additions or corrections. Authors should pay attention to scientific names of eel, also phrase between lines 261-267 should be clearly written if it refers to the human population.   Chapter 4. Results. (Chapter 2 in presented paper)The results are well presented. Chapter 4. does not require additions or changes.   Chapter 5. Conclusions.   The Conclusions chapter is well presented. It does not require additions or changes.

References are nearly appropiate. The number of references can be adjusted to keep the newest papers with the highest references related to the paper.

Author Response

Abstract: The abstract is well presented. Chapter does not require vital additions or changes. For a better understanding authors should give scientific names of eel and Chinese hamster (rows 15-16).   Keywords expresses the content of the work.   

We changed “eel and Chinese hamster ovary DG44” to “Anguillid eel” and “Cricetulus griseus ovary DG44”.

We changed Keywords by reviewer’s comments

Chapter 1. Introduction. The introduction chapter presents the importance of hormones, a description of them as well as results on recombinant hormone in Chinese hamster ovary cells. 

Authors should give scientific names of Chinese hamster (row 50), eel (row 64)   

→We inserted “the Cricetulus griseus” in the Line

→We inserted “Anguillid” in the Line

Chapter 2. Materials and Methods.    The Materials and methods chapter is composed of twelve subchapters as follows: (new numbered)

2.1. Materials 

2.2. Construction of eel FSH-wt and FSH-M vectors 

2.3. Transfection into CHO DG44 cell 

2.4. Isolation of single cells expressing rec-eel FSH-wt and FSH-M proteins 

2.5. Production of rec-eel FSH-wt and FSH-M proteins 

2.6. ELISA for rec-eel FSH protein quantitation 

2.7. Western blotting analysis of rec-eel FSH proteins 

2.8. Enzymatic release of N-linked oligosaccharides 

2.9. Analysis of cAMP levels by homogenous time-resolved fluorescence assays 

2.10. Measurement of pERK1/2 levels by homogenous time-resolved fluorescence assays 

2.11. Analysis of phospho-ERK1/2 activation by western blot 

2.12. Data Analysis 

Authors should change in paper the order of chapters Chapter 4 to become Chapter 2 and vice versa, and also the 12 sub-chapters to be summarized in such a way as to present clearly the materials and methods used.   

→We arranged by the Manuscript submission & instructions for Intl J Mol Sci Journal authors  

Chapter 3. Discussion. Chapter 3 Discussions is well-proportioned and makes the necessary and informative references on the research results. 

Chapter 3. does not require additions or corrections. Authors should pay attention to scientific names of eel, also phrase between lines 261-267 should be clearly written if it refers to the human population.   

We inserted “adult’s human” in the Line 273.

We changed “Therefore, eel FSH-M could represent a longer-acting formulation; however, in vivo experiments are needed to evaluate this “to Although the in vivo experiment was not conducted in the present study, rec-eel FSH-M could represent a longer-acting formulation. Therefore, the in vivo experiments are needed to evaluate this.

Chapter 4. Results. (Chapter 2 in presented paper) The results are well presented. Chapter 4. does not require additions or changes.   

We arranged by the Manuscript submission & instructions for Intl J Mol Sci Journal authors 

Chapter 5. Conclusions.   The Conclusions chapter is well presented. It does not require additions or changes.

References are nearly appropiate. The number of references can be adjusted to keep the newest papers with the highest references related to the paper.

 We checked conclusions.

Round 2

Reviewer 2 Report

Comments and Suggestions for Authors

The authors appropriately responded to the comments raised by the reviewer

Comments on the Quality of English Language

Minor editing of English language required

Author Response

Reviewer 2

Comments and Suggestions for Authors

The authors appropriately responded to the comments raised by the reviewer

Comments on the Quality of English Language

Minor editing of English language required

→ The manuscripts reedited by Special English Editing company by reviewer’s comment.

Reviewer 3 Report

Comments and Suggestions for Authors

Re. IJMS 3027470

 Article 

Stable Production of a Recombinant Single-Chain Eel Follicle-Stimulating Hormone Analog in CHO DG44 Cells 

 Munkhzaya Byambaragchaa 1,2, Sei Hyen Park 3, Sang-Gwon Kim 3, Min Gyu Shin 4, Shin-Kwon Kim 4, Myung-Hum Park 5, Myung-Hwa Kang and Kwan-Sik Min 1,2,7,

The authors must present the scientific paper following the order of the chapters so that the Materials and Methods Chapter must become Chapter 2 and not Chapter 4. 

Also in the Materials and Methods Chapter (in each of sub chapter) the authors must present much more clearly the materials and methods used.

Author Response

Reviewer 3

The authors must present the scientific paper following the order of the chapters so that the Materials and Methods Chapter must become Chapter 2 and not Chapter 4. 

 The manuscript was followed by the author’s procedure.

Also in the Materials and Methods Chapter (in each of sub chapter) the authors must present much more clearly the materials and methods used.

 We tried my best to express it in detail. The changed contents were marked in blue.
